# A Novel Microstructural Evolution Model for Growth of Ultra-Fine Al_2_O_3_ Oxides from SiO_2_ Silica Ceramic Decomposition during Self-Propagated High-Temperature Synthesis

**DOI:** 10.3390/ma13122821

**Published:** 2020-06-23

**Authors:** Mateusz Kopec, Stanisław Jóźwiak, Zbigniew L. Kowalewski

**Affiliations:** 1Institute of Fundamental Technological Research, Polish Academy of Sciences, Pawińskiego 5B, 02-106 Warszawa, Poland; zkowalew@ippt.pan.pl; 2Faculty of Advanced Technologies and Chemistry, Military University of Technology, 00-908 Warsaw 49, Poland; Stanislaw.jozwiak@wat.edu.pl

**Keywords:** intermetallics, powder methods, electron microscopy, X-ray analysis

## Abstract

In this paper, experimental verification of the microstructural evolution model during sintering of aluminum, iron and particulate mullite ceramic powders using self-propagated high-temperature synthesis (SHS) was performed. The powder mixture with 20% wt. content of reinforcing ceramic was investigated throughout this research. The mixed powders were cold pressed and sintered in a vacuum at 1030 °C. The SHS reaction between sintered feed powders resulted in a rapid temperature increase from the heat generated. The temperature increase led to the melting of an aluminum-based metallic liquid. The metallic liquid infiltrated the porous SiO_2_ ceramics. Silicon atoms were transited into the intermetallic iron–aluminum matrix. Subsequently, a ternary matrix from the Fe–Al–Si system was formed, and synthesis of the oxygen and aluminum occurred. Synthesis of both these elements resulted in formation of new, fine Al_2_O_3_ precipitates in the volume of matrix. The proposed microstructural evolution model for growth of ultra-fine Al_2_O_3_ oxides from SiO_2_ silica ceramic decomposition during SHS was successfully verified through scanning electron microscopy (SEM), X-ray energy-dispersive spectroscopy (EDS) analysis and X-ray diffraction (XRD).

## 1. Introduction

Intermetallic–ceramic composites (IMCs) are a narrow group of composites used as structural and functional materials. These composites combine unique properties of ceramics (hardness, thermo-chemical resistance and thermal stability) with properties of metals, i.e., mechanical strength. Due to their superior properties, there is a wide spectrum of possible applications for these materials [1]. Composites, in comparison to conventional structural materials, are characterized by a higher Young’s modulus, a low coefficient of thermal expansion, abrasion resistance and high strength. These properties could be also maintained during operation at elevated temperatures [2].

The potential applications of IMC materials have led to a growing number of publications describing their properties and manufacturing technologies. A number of studies investigating the manufacturing processes of IMCs reinforced by iron, aluminum and oxide particles with various mass contents of strengthening elements—Fe (15%–93%), Al (5%–70%) and Si (0.5%–25%)—are found in the literature [3,4,5,6,7,8].

The review of IMC manufacturing techniques presents a wide spectrum of possibilities for IMC fabrication from conventional sintering methods [4] to novel and advanced techniques: for example, diffusion bonding of plates made of high-purity input materials [5,6,7]. For commercial applications, it is possible to obtain the final product by using a conventional sintering method, reactive sintering [9] or advanced self-propagated high-temperature synthesis (SHS) [3,4,10]. An important aspect in manufacturing high-performance IMCs is the use of the finest possible reinforcing phase powders (<1 µm) during the sintering process. By using such powders, a relatively good distribution of reinforcement in the matrix volume and its location in the sinter volume can be achieved. Fine dimensions of reinforcement powders might be obtained by using high-energy milling techniques [11,12,13,14,15]. Satisfactory effects of powder fragmentation have been observed after a long milling time of over 5 h [16,17]. A long milling time includes technological breaks for stabilization of milling conditions and results in a long processing time and low efficiency of the whole process. In the wide spectrum of IMCs, Al_2_O_3_ reinforced composites are characterized by the best mechanical properties [18,19,20]. They are usually reinforced with dispersive Al_2_O_3_ oxides obtained during synthesis or thermite reaction [20,21,22,23,24,25,26,27].

Different approaches to model the sintering processes and microstructure evolutions are reported in the literature. These include phenomenological and mechanistic methods, continuous and discrete formulations, modeling at both macro and micro levels, as well as multiscale modeling [28,29,30,31,32]. Other methods include microstructural evolution models. An interesting approach reported in [33] allows for development of a phase-field model to investigate intermetallic compound phase transformation and nucleation of one of the phases involved. On the one hand, phase-field methods are considered as powerful tools for microstructure modeling [34]. They could be used to model various processes, including rapid solidification [35], additive manufacturing [36] and various materials [37,38]. On the other hand, new studies on microstructural evolution during heating of carbon nanotube metal matrix composites (CNT-MMC) with additional modeling attempts have been recently reported [39]. The variety of materials and composites extends the possibilities of microstructure evolution modeling to different methods and approaches. 

Based on the detailed literature review, a novel microstructure evolution model for investigating powder mixtures under high-temperature sintering was developed. The rapid temperature increase, generated during SHS, led to the melting of aluminum-based metallic liquid. The metallic liquid infiltrated the porous SiO_2_ ceramics, and silicon atoms were transited into the intermetallic iron–aluminum matrix. Subsequently, formation of an Fe–Al–Si ternary matrix and synthesis of the oxygen and aluminum occurred. Synthesis of both these elements resulted in formation of new, fine Al_2_O_3_ precipitates in the volume of matrix. A model for growth of ultra-fine Al_2_O_3_ oxides from decomposition of SiO_2_ silica during self-propagated high-temperature synthesis was experimentally verified through vacuum sintering, combined with high-energy milling of reinforcement particles. This method allowed for obtainment of a permanent connection between matrix and reinforcement in the fabricated IMC/Al_2_O_3_ composite. Additionally, high-energy powder milling of the reinforcing phase promises a reduction of the composite inhomogeneity after high-temperature sintering.

## 2. Materials and Methods 

The powder mixture of 28% wt. of iron and 52 wt.% of aluminum with 20 wt.% of the reinforcing mullite ceramic was proposed to evaluate the microstructure evolution model. The specific contents of aluminum and iron were selected to obtain an intermetallic matrix of the sintered material. Milling parameters were selected on the basis of literature analysis, the high-energy milling machine manufacturer’s recommendations and the authors’ experience. A grinding jar and milling balls of 10 mm diameter were made of 100 Cr6 steel. The milling balls mass to feed powder mass ratio was equal to 10:1. The powder mixture was grinded for 5 min with a rotating speed of 450 RPM. After initial grinding, the mixture of mullite, iron and aluminum powders was put into a turbulent mixer for 30 min. The mixed powders were then pressed in a 25-mm-diameter die in a single-sided hydraulic press at 900 MPa for 3 min. The compacts obtained were located inside the graphite dies and sintered in a vacuum chamber at 1030 °C for 5 min without external load. The microstructural characterization was performed on an FEI Scios field emission gun scanning electron microscope (FEG-SEM) operated at 20 kV. Prior to this study, the specimens were first hot mounted and then ground using 80, 180, 300, 600, 800, 1200 and 4000 SiC paper. The polishing was performed using Metrep^®^ MD-Chem cloth with 3 µm diamond suspension. The XRD measurements were performed using a Rigaku (Tokyo, Japan) Ultima IV diffractometer with Co-K radiation (λ¼1.78897 Å) and operating parameters of 40 mA and 40 kV with a scanning speed of 1°/min and a scanning step of 0.02° in the range of 20°–120°.

## 3. Introduction of Microstructure Evolution Model for Growth of Ultra-Fine Al_2_O_3_ Oxides from SiO_2_ Silica Ceramic Decomposition

Literature analysis allowed for proposal of a model for microstructural evolution during high-temperature sintering of IMCs by using powder metallurgy technology. This model includes the sintering process of iron, aluminum and mullite ceramics, and simultaneous formation of oxide precipitates formed from the defragmentation of SiO_2_ during the infiltration process. The model is mainly based on the SHS reaction between sintered feed powders (Figure 1a), resulting in a rapid temperature increase from the heat generated during the SHS. The temperature increase led to the melting of aluminum-based metallic liquid. The metallic liquid infiltrated the porous ceramics, which led to their defragmentation (Figure 1b). The mechanism of the oxide precipitates formation on the basis of SiO_2_ decomposition can be represented by the following relationship:SiO_2_ + Fe_x_Al_x_ → Fe_x_Al_x_Si_x_ + Al_2_O_3_(1)

Mullite ceramics exhibit wettability of liquid aluminum at 1000 °C [40], and thus an infiltration process may accelerate their defragmentation. During the sintering process, silicon atoms are expected to be transited into the intermetallic iron–aluminum matrix. Such a transition leads to the formation of a ternary matrix from the Fe–Al–Si system and subsequent synthesis of the oxygen and aluminum. Synthesis of both these elements results in formation of new, fine Al_2_O_3_ precipitates in the volume of matrix (Figure 1c) within the primary areas of silica ceramics. Finally, an IMC composite with a permanent connection between the ceramics and the intermetallic matrix can be obtained.

## 4. Results and Discussion

### 4.1. Optimization of Sintering Temperature for Investigated Powder Mixtures

The initial sintering parameters were selected for the temperature of 1200 °C, compressing load of 25 kN and sintering time of 5 min. However, these parameters did not allow for obtainment of a homogeneous structure of sintered material. During the sintering process, the pressure inside the graphite dies was so high that it caused them to crack. Therefore, in subsequent stages, sintering without external load was proposed. Unfortunately, during the process, liquid matrix material leaked between the graphite die and the upper punch pressing the sintered compact. Leaked material was characterized by using scanning electron microscopy (SEM) and chemical composition analysis (EDS), as presented in Figure 2 and Table 1. It was found that the contents of silicon and aluminum within the microstructure of material tested (Figure 2) were comparable to the hypoeutectic silumin characterized by the melting point of 577 °C. In order to prevent material loss during the heating stage, the sintering parameters were modified accordingly. It was found that an absence of metallic liquid leaks was obtained during sintering without external load and subsequent reduction of the sintering temperature. The sintering temperature was reduced to 1030 °C. Relatively small but obvious differences in the sintering temperature were caused by a change in the silicon content in the material structure. The higher content of silicon corresponded to the lower value of melting temperature. An application of new sintering parameters allowed for fabrication of the sinters, without any material losses, and moreover for changes in the chemical composition of the material.

In order to obtain a homogeneous phase structure of the matrix, it was necessary to perform a homogenization process. It is widely accepted that in order to have more effective homogenization, a higher temperature in the process should be used. Higher values of temperature accelerate atom diffusion and subsequent phase transformations. The temperature was selected based on the analysis of the ternary aluminum–iron–silicon equilibrium system [41]. The temperature of 900 °C was selected in order to avoid the melting point identified on the basis of the chemical composition analysis of the phases formed after the sintering process. The sintering time of 5 min was used on the basis of the authors’ research [42], where the optimized structure and properties of FeAl intermetallic phases were obtained.

### 4.2. Microstructural Characterization of Sintered Material and Experimental Verification of the Model

A microstructure of the sintered sample with 20% wt. reinforcement was characterized by the large, nonfragmented areas of the primary oxide precipitates in a ternary iron–aluminum–silicon matrix (Figure 3). The oxide ceramics were evenly distributed in the volume of the material, as presented in Figure 3a. The material matrix consisted of the FeAl_3_ and Al_4.5_FeSi (τ_6_) phases (Figure 3b). The τ_6_ phase was mainly observed in the areas close to the ceramic reinforcement.

The correctness of the model assumed was demonstrated by microanalysis of the chemical composition performed in the area of primary defragmented SiO_2_ porous ceramic particles (Figure 3c,d and Table 2). It was observed that the dark areas (“3”) were characterized with an increased proportion of oxygen and aluminum in comparison to the light areas (“1”, “2”), indicating the dominant content of alundum ceramics. The light gray areas were characterized by increased contents of silicon and iron. The contents of silicon and iron in the material matrix confirmed that these specific zones were formed by the transition of silicon atoms from SiO_2_ ceramics to the intermetallic matrix.

According to the microstructure evolution model and microstructural observations, phase transformations were described by the following equation:FeAl_3_ + SiO_2_ → Al_4.5_FeSi (τ_6_) + Al_2_O_3_(2)

One can indicate that high-energy milling led to defragmentation of mullite ceramics to Al_2_O_3_ conglomerates mechanically bonded with SiO_2_ of an initial size equal to ~20 μm (Figure 4a). A detailed analysis of the primary oxide ceramics areas revealed the initiation of metallic liquid infiltration into the porous ceramics. During sintering at 1030 °C, the defragmentation process of primary SiO_2_ precipitates was observed, as presented in Figure 4b. Since the liquid metal infiltrated the porous silica ceramic, the silicon atoms diffused into the FeAl_3_ phase. The primary two-component matrix was transformed into the ternary Al_4.5_FeSi (τ_6_) phase (Figure 4c). Additionally, the oxygen atoms diffused from SiO_2_ ceramics and reacted with the most chemically active aluminum atoms taken from the metallic liquid. As a consequence, fine, spherical Al_2_O_3_ particles (with diameters less than 1 μm) in the iron–aluminum–silicon matrix were formed (Figure 4d). The mechanism observed for growth of new, fine Al_2_O_3_ aluminum oxide precipitates confirmed the correctness of the microstructure evolution model.

The SHS reaction provided a large amount of energy and subsequent immediate temperature increase. The temperature increase led to melting of the FeAl_3_ matrix and its transformation to Al_4.5_FeSi (τ6). Such material behavior during the SHS process, where the external pressure during sintering was not provided, may lead to the formation of porous structures, as reported in [43]. Subsequent homogenization should accelerate the densification process and, as a consequence, increase the density of the material.

### 4.3. X-ray Phase Analysis of Sintered Material

The assumptions of the material model were subsequently confirmed by using in situ X-ray phase analysis. The diffractograms were recorded at the actual test temperature ranging from 25 to 900 °C (Figure 5). X-ray in situ analysis was performed by using a unique, high-temperature HTK attachment. It allowed for phase structure analysis as the function of time and temperature. In order to ensure the correctness of temperature selection, the sample was annealed for 1 h before each in situ X-ray diffraction measurement. The following phase transformations were observed:
At temperatures up to 500 °C, the material mainly consisted of Al_2_O_3_, SiO_2_, Fe_2_Al_9_ and Al_2_FeSi phases. Such phase compositions indicated that the temperature was too low to initiate phase transformations.In the temperature range of 500 to 600 °C, the material mainly consisted of the Al_2_O_3_ phase and Al_2_Fe_3_Si_3_ and Fe_2_Al_9_ phases, which were presumably indirect products of the phase transformations initiated. At the temperature of 700 °C, an intense peak of Al_2_O_3_ was observed. An occurrence of such a peak was associated with the formation of new, fine precipitates of this oxide.At temperatures higher than 700 °C, the sinter phase composition stabilized. It consisted of the Al_4.5_FeSi phase and new Al_2_O_3_. The homogenization at the temperature of 900 °C did not affect the phase structure of the material significantly because only one additional peak of the Al_2_Fe_3_Si_3_ phase was observed.The Halder–Wagner (HW) method was used to determine crystallite size in the function of temperature (Figure 6). The size of Al_2_O_3_ crystallites increased with the temperature, and the average value of 392 Å was obtained after heating to 900 °C.

## 5. Conclusions

The performed microstructural characterization and further analysis confirmed the correctness of the proposed model, aimed primarily at obtaining fine precipitates of Al_2_O_3_ oxide reinforcement in the intermetallic matrix of the composite. However, during the classical sintering process in the graphite dies using external pressure, the main problems were associated with the transition of silicon into the metallic liquid. This phenomenon led to an increase of the metallic liquid phase fluidity. Such material behavior combined with external pressure and caused loss of material during the sintering process. It was proposed, therefore, to perform the sintering process without external pressure. The improved sintering process allowed for successful formation of the composite material. Moreover, the occurrence of the assumed phase transformations was observed, in particular the defragmentation of SiO_2_ oxide and the construction in its primary areas of fine, spherical Al_2_O_3_ oxide precipitations, arranged in a ternary intermetallic iron–aluminum–silicon matrix. Based on the studies performed showing the correctness of the proposed model, one can recommend it for further usage in manufacturing of IMC composites. It should be emphasized here that the application of external loads during the sintering process to prevent the outflow of metallic liquid from infiltrating ceramics is crucial and must be taken into account. It was concluded that a technology potentially capable of ensuring successful sintering of IMCs could be hot isostatic pressing (HIP).

Summing up the results of this work, one can state in particular:
The proposed model of composite fabrication, based on the decomposition of mullite ceramics infiltrated with a metallic liquid, allows for obtainment of a composite material with an intermetallic matrix reinforced with Al_2_O_3_ ceramic particles derived from the defragmentation of primary alundum ceramics and newly formed fine-grained spherical ceramics resulting from SiO_2_ decomposition.The parameters of the sintering process and subsequent homogenization are significantly affected by the participation of silicon, which significantly increases the fluidity of the metallic liquid formed during sintering and the final phase structure of the sintered matrix.The correctness of the assumed model must be supported by using proper sintering conditions that would lead to satisfactory cohesion between the matrix and reinforcement. Future promising sintering technology seems to be hot isostatic pressing that should guarantee fabrication of IMC composites with satisfactory performance.

## Figures and Tables

**Figure 1 materials-13-02821-f001:**
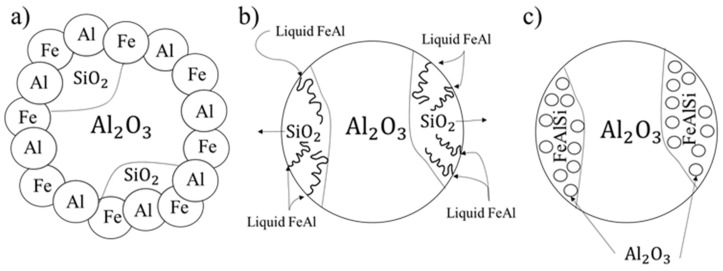
Schematic model of microstructure evolution during the self-propagated high-temperature synthesis (SHS) reaction: (**a**) initiation of the SHS reaction between sintered feed powders, (**b**) defragmentation of porous ceramics through infiltration, and (**c**) formation of new, fine Al_2_O_3_ precipitates in the volume of matrix.

**Figure 2 materials-13-02821-f002:**
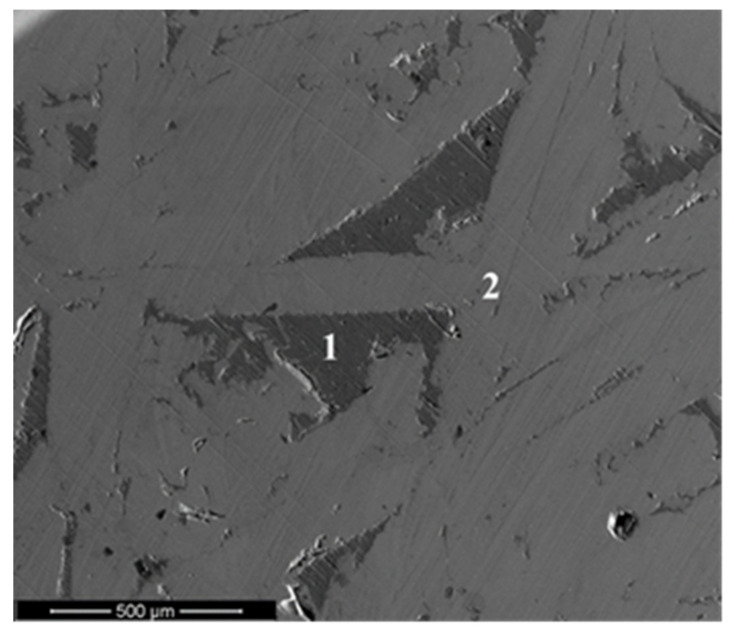
Microstructure of material leaked during sintering at 1200 °C.

**Figure 3 materials-13-02821-f003:**
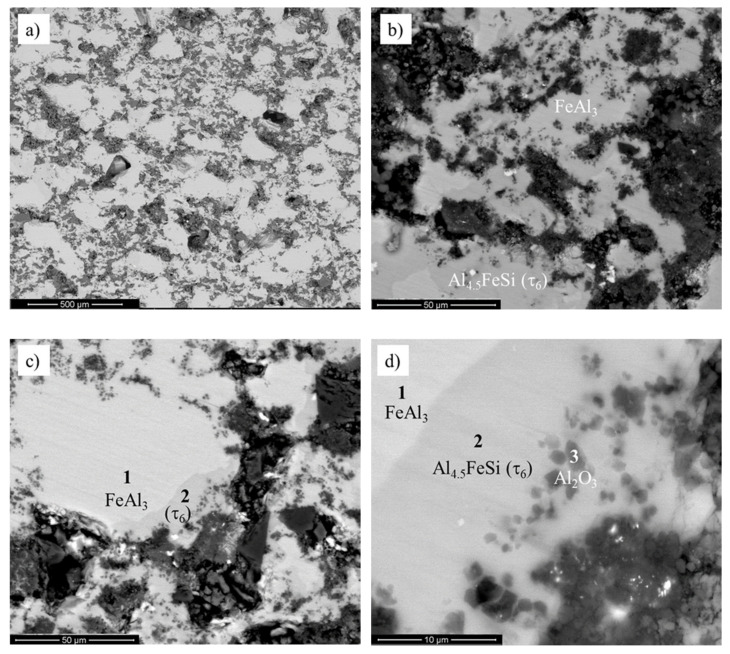
Microstructure of composite with 20% wt. content of mullite ceramics after sintering at 1030 °C for 5 min (**a**); FeAl_3_ + Al_4.5_FeSi (τ_6_) multiphase matrix (**b**); an initiation of the disintegration process of primary SiO_2_ precipitates (**c**); silicon diffusion into the intermetallic iron–aluminum matrix and newly formed Al_2_O_3_ ceramics in the areas of primary SiO_2_ particles (**d**).

**Figure 4 materials-13-02821-f004:**
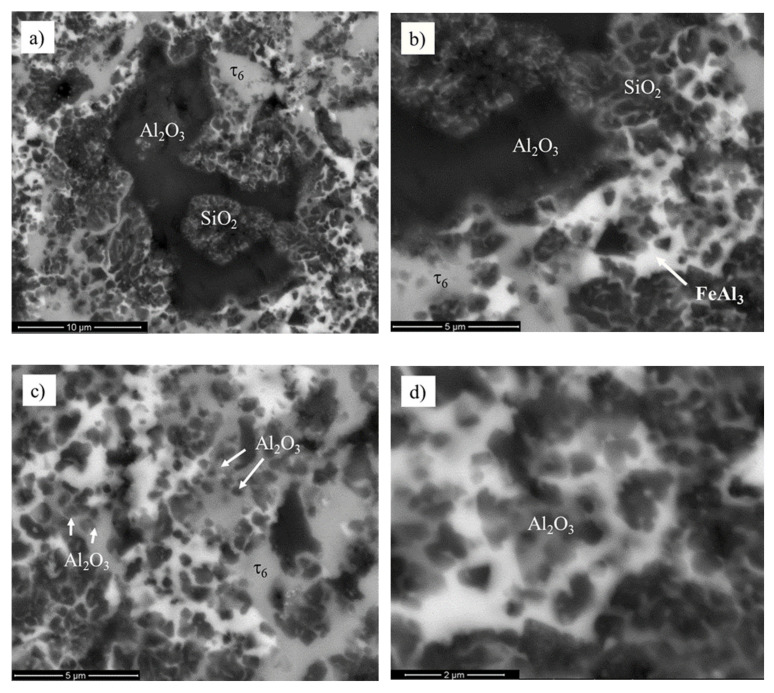
Mechanically bonded Al_2_O_3_/SiO_2_ conglomerates (**a**); defragmentation of the silica ceramics through infiltration (**b**); growth of fine Al_2_O_3_ oxides in τ_6_ matrix (**c**); structure of spherical Al_2_O_3_ particles within the matrix (**d**).

**Figure 5 materials-13-02821-f005:**
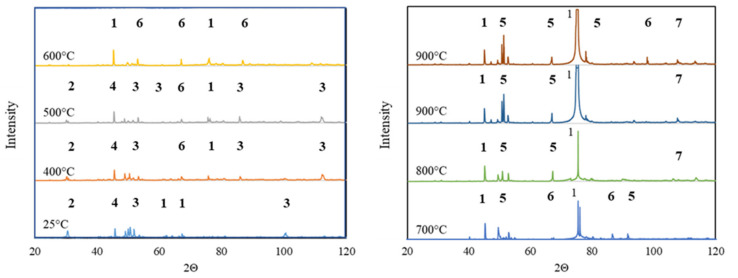
Phase structure evolution during heating from 25 to 900 °C, where 1—Al_2_O_3_, 2—SiO_2_, 3—Fe_2_Al_9_, 4—Al_2_FeSi, 5—Al_4.5_FeSi, 6—Al_2_Fe_3_Si_3_ and 7—FeAl_3_.

**Figure 6 materials-13-02821-f006:**
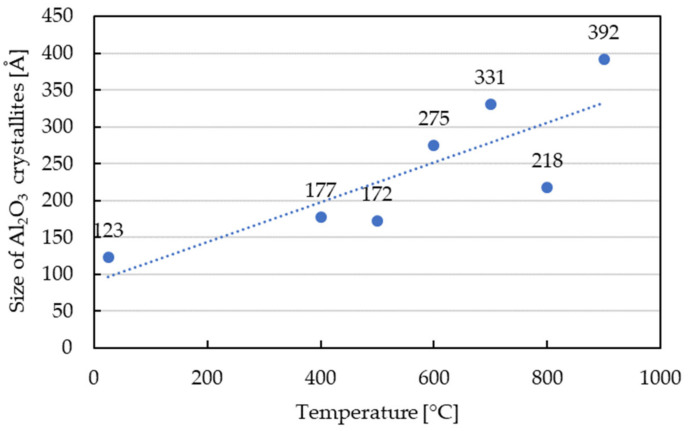
Evolution of Al_2_O_3_ crystallites size during heating.

**Table 1 materials-13-02821-t001:** Chemical composition of material leaked during sintering at 1200 °C.

Wt.%	Al	Si	Fe
**1**	87.95	8.92	3.13
**2**	76.18	1.91	21.94

**Table 2 materials-13-02821-t002:** Chemical composition of the decomposed area.

Wt.%	Al	Si	Fe	O
**1**	62.39	2.93	34.68	-
**2**	55.69	14.98	29.33	-
**3**	61.27	-	8.73	30.00

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
