# Peer review of "A Novel Microstructural Evolution Model for Growth of Ultra-Fine Al2O3 Oxides from SiO2 Silica Ceramic Decomposition during Self-Propagated High-Temperature Synthesis"

_materials, 2020, doi:10.3390/ma13122821_

Round 1

Reviewer 1 Report

General comment In the current paper, authors have developed model for growth of ultra-fine Al2O3 from SiO2 decomposition during the SHS processTo publish the paper, the authors should majorly revise the paper and extend it. I have listed my comments on the paper below. 1. I strongly suggest that the authors explain more about the microstructure evolution model in the abstract and the last paragraph of the introduction section. 2. I strongly suggest that the authors present a more detailed literature review on the previous microstructure evolution model in the introduction section. 3. The authors jump into the results and discussion too early without elaborating enough on the methodology used. 4. It would be better if section 3.1 is presented before results and discussion. 5. Line 81 of Page 2 refers to literature analysis and previous research without citing a single reference. I strongly suggest that the authors add some citations and elaborate more deeply. 6. There are two section 3.1 in the manuscript. 7. Conclusion must be much more comprehensive, summarising the methodology.

Author Response

Detailed Response to Reviewer Comments

Ms. Ref. No.: materials-832120

Title: A Novel Microstructural Evolution Model for Growth of Ultra-Fine Al2O3 Oxides from SiO2 Silica Ceramic Decomposition During Self-Propagated High Temperature Synthesis

Materials Dear Sir or Madame,  

I would like to thank you very much for your letter and the reviewer’s comments on our manuscript (No.: materials-832120). We appreciate your very valuable comments, that gave us a  chance for revising the manuscript.

We have addressed all of the comments and revised the manuscript accordingly. All of the changes have been highlighted in yellow in the revised manuscript. Detailed responses to the comments are described in the “Response to Reviewers” point by point.

We now resubmit the manuscript for your further consideration for publication in your journal. We sincerely hope this revised manuscript will be finally acceptable for publication. If you have any questions about this manuscript, please do not hesitate to contact me.

Best regards

Mateusz Kopec

On behalf of all co-authors

Institute of Fundamental Technological Research

Polish Academy of Sciences

Reviewer’s Comments:

Reviewer #1:

General comment In the current paper, authors have developed model for growth of ultra-fine Al2O3 from SiO2 decomposition during the SHS process. To publish the paper, the authors should majorly revise the paper and extend it. I have listed my comments on the paper below.

  1. I strongly suggest that the authors explain more about the microstructure evolution model in the abstract and the last paragraph of the introduction section.

Response: We would like to thank the reviewer for the comment. Corrections were highlighted in the article and detailed response is located below.

Abstract: SHS reaction between sintered feed powders resulted in a rapid temperature increase from the heat generated. The temperature increase leads to the melting of aluminium based metallic liquid. A metallic liquid infiltrated the porous SiO2 ceramics. Silicon atoms were transited into the intermetallic iron-aluminium matrix. Subsequently, a ternary matrix from the Fe-Al-Si system was formed, and synthesis of the oxygen and aluminium occurred. A synthesis of both these elements results in a formation of new, fine Al2O3 precipitates in the volume of matrix. (Lines 17 - 23)

Introduction: Different approaches to model the sintering processes and microstructure evolutions were found in the literature. These approaches include phenomenological and mechanistic methods, continuous and discrete formulations, and modelling at both, macro and micro level as well as multi scale modelling [28-32]. Other methods include the microstructural evolution models. Interesting approach reported in [33] allows to develop phase-field model to investigate intermetallic compound phase transformation and nucleation of one of the phases involved. On the one hand, phase-field methods are considered as powerful tools for microstructure modelling [34]. They could be used to model the various processes including rapid solidification [35] or additive manufacturing [36] and various materials [37, 38]. On the other hand, new studies on microstructural evolution during heating of CNT/Metal Matrix composites with additional modelling attempts were recently  reported [39]. The variety of materials and composites extend the possibilities of microstructure evolution modelling to different methods and approaches.

Based on the detailed literature review, a novel microstructure evolution model for investigated powder mixtures under high temperature sintering was developed. The rapid temperature increase, generated during SHS, leads to the melting of aluminium based metallic liquid. A metallic liquid infiltrated the porous SiO2 ceramics and silicon atoms were transited into the intermetallic iron-aluminium matrix. Subsequently, a formation of Fe-Al-Si ternary matrix and synthesis of the oxygen and aluminium occurred. A synthesis of both these elements results in a formation of new, fine Al2O3 precipitates in the volume of matrix. (Lines 57 – 75)

  1. I strongly suggest that the authors present a more detailed literature review on the previous microstructure evolution model in the introduction section.

Response: We would like to thank the reviewer for the comment. Corrections were highlighted in the article and detailed response is located below.

Different approaches to model the sintering processes and microstructure evolutions were found in the literature. These include phenomenological and mechanistic methods, continuous and discrete formulations, and modelling at both, macro and micro level as well as multi scale modelling [28-32]. Other methods include the microstructural evolution models. Interesting approach reported in [33] allows to develop phase-field model to investigate intermetallic compound phase transformation and nucleation of one of the phases involved. On the one hand, phase-field methods are considered as powerful tools for microstructure modelling [34]. They could be used to model the various processes including rapid solidification [35] or additive manufacturing [36] and various materials [37, 38]. On the other hand, new studies on microstructural evolution during heating of CNT/Metal Matrix composites with additional modelling attempts were recently  reported [39]. The variety of materials and composites extend the possibilities of microstructure evolution modelling to different methods and approaches. (lines 57 – 75)

  1. The authors jump into the results and discussion too early without elaborating enough on the methodology used.

Response: We would like to thank the reviewer for the comment. In authors opinion, a sufficient information about methodology was presented in Materials and Methods section (Lines 82 – 98). Authors provided each single step of experimental procedure carried out. The content of feed powders with explanation was provided in Lines 82 – 85. Subsequently, the powder milling and mixing parameters were provided in Lines 87 – 90 including the most important parameters:

  • grinding jar and milling balls material,
  • milling balls mass to feed powder mass ratio,
  • grinding time,
  • rotating speed,
  • mixing time.

Such treated powder mixture was then cold pressed using 25mm diameter die in a single-sided hydraulic press at 900MPa for 3 min. and compacts were sintered inside the graphite dies a vacuum chamber at 1030°C for 5 minutes without external load as reported in Lines 91-93. At the end of paragraph, brands of microscope and diffractometer with operational parameters were presented. If any further information are needed, authors would be more than happy to provide them.

  1. It would be better if section 3.1 is presented before results and discussion.

Response: We would like to thank the reviewer for the comment. Subsection 3.1 was separated and new Section 3 was provided and highlighted (Line 101). Additionally, section 4. Results and Discussion was divided into three subsections as highlighted in manuscript. (Lines 123, 158 and 207).

  1. Line 81 of Page 2 refers to literature analysis and previous research without citing a single reference. I strongly suggest that the authors add some citations and elaborate more deeply.

Response: We would like to thank the reviewer for the comment. The literature analysis on modelling methods was provided according to second comment of reviewer.

  1. There are two section 3.1 in the manuscript.

Response: We would like to thank the reviewer for the comment. According to 4th comment, subsection 3.1 was separated and new section 3 was provided and highlighted. Also, section 4. Results and Discussion was divided into three subsection as highlighted in manuscript. (Lines 123, 158 and 207).

  1. Conclusion must be much more comprehensive, summarising the methodology.

Response: We would like to thank the reviewer for the comment. An adequate modification was highlighted in the article (Lines 236 – 242) and detailed response is presented below.

A correctness of the novel microstructural evolution model for growth of the ultra-fine Al2O3 oxides from decomposition of the SiO2 silica ceramic during self-propagated high temperature synthesis was successfully verified via the experiments. Vacuum sintering of feed powders at 1030°C allowed to obtain a ternary Al4.5FeSi (τ6) matrix reinforced with new, fine grain Al2O3 particles obtained from the decomposition of SiO2 ceramics. The process of infiltration of porous SiO2 ceramic was observed through SEM method. Additional EDS and XRD analysis confirmed the phase transformations assumed in proposed microstructural evolution model.

Reviewer 2 Report

The paper displays the fabrication of a type of IMC, although the authors call it ICM on the second page. This could be very interesting to the community after major revision is carried out. 

1, Please comment on the unique properties (mechanical or functional) of this composite. 

2, Please explain how the sintering condition is determined, especially the dwell time which is very short. 

3, Mullite ceramic only takes 20 wt% in the raw materials. Why does Al2O3 seem to occupy the major portion of the model in Fig. 1?

4, What are the very dark contrast in Fig. 2? It seems to me that those big clusters instead of the "ultrfine" Al2O3 particles, are the global feature of the obtained composite. Does it imply that the intended microstructure is not successfully obtained?

5, I don't get how the SEM micrographs reveal the "infiltration" process. Please elaborate in detail, since that is the essential part of the model. 

6, Please calculate the crystallite size of Al2O3 particles based on the XRD pattern.

7, Is the in situ XRD conducted on a hot stage? I suppose so. Then why is that necessary? In other word, why not collect the pattern at room temperature in the sample which is cooled down from different annealing conditions? Or, do the author believe that there will be no evolution above 900C?

8, Please comment on the densification process, in addition to the chemical reaction. 

Author Response

Detailed Response to Reviewer Comments

Ms. Ref. No.: materials-832120

Title: A Novel Microstructural Evolution Model for Growth of Ultra-Fine Al2O3 Oxides from SiO2 Silica Ceramic Decomposition During Self-Propagated High Temperature Synthesis

Materials Dear Sir or Madame,  

I would like to thank you very much for your letter and the reviewer’s comments on our manuscript (No.: materials-832120). We appreciate your very valuable comments, that gave us a  chance for revising the manuscript.

We have addressed all of the comments and revised the manuscript accordingly. All of the changes have been highlighted in yellow in the revised manuscript. Detailed responses to the comments are described in the “Response to Reviewers” point by point.

We now resubmit the manuscript for your further consideration for publication in your journal. We sincerely hope this revised manuscript will be finally acceptable for publication. If you have any questions about this manuscript, please do not hesitate to contact me.

Best regards

Mateusz Kopec

On behalf of all co-authors

Institute of Fundamental Technological Research

Polish Academy of Sciences

Reviewer’s Comments:

Reviewer #2:

The paper displays the fabrication of a type of IMC, although the authors call it ICM on the second page. This could be very interesting to the community after major revision is carried out.

  1. Please comment on the unique properties (mechanical or functional) of this composite.

Response: We would like to thank the reviewer for the comment. Mechanical and functional properties of this composite was not studied as yet. Authors are planning to perform some preliminary tests (uniaxial tension test,  heat and wear resistant test) in the future and present them in different paper.

  1. Please explain how the sintering condition is determined, especially the dwell time which is very short.

Response: We would like to thank the reviewer for the comment. The justification of sintering conditions was highlighted in manuscript as new subsection ‘Optimization of sintering temperature for investigated powder mixtures’ (Lines 123 – 157). It is also presented below.

The initial sintering parameters were selected for the temperature of 1200°C, compressing load of 25kN and sintering time of 5 min. However, these parameters did not allow to obtain a homogeneous structure of sintered material. During sintering process, the pressure inside the graphite dies was so high that caused its crack. Therefore in subsequent stages, sintering without external load was proposed. Unfortunately, during the process liquid matrix material leaked between the graphite die and the upper punch pressing the sintered compact. Leaked material was characterized by using a Scanning Electron Microscopy (SEM) and chemical composition analysis (EDS) as presented in Figure 2 and Table 1. It was found, that the content of silicon and aluminum within the microstructure of material tested (Figure 2) was comparable to the hypoeutectic silumin characterized by the melting point of 577°C. In order to prevent a material loss during the heating stage, the sintering parameters were modified accordingly. It was found that an absence of the metallic liquid leaks was obtained during sintering without external load and subsequent reduction of the sintering temperature. The sintering temperature was reduced to 1030°C. Relatively small, but obvious differences in the sintering temperature were caused by a change of the silicon content in the material structure. The higher content of silicon, corresponded to the lower value of melting temperature. An application of new sintering parameters allowed to fabricate the sinters without any material losses, and moreover to change the chemical composition of the material.

Figure 2. Microstructure of material leaked during sintering at 1200°C.

Table 1. Chemical composition of material leaked during sintering at 1200°C.

Wt.%

Al

Si

Fe

1

87.95

8.92

3.13

2

76.18

1.91

21.94

In order to obtain a homogeneous phase structure of the matrix, it is necessary to perform a homogenization process. It is widely accepted that in order to have a more effective homogenization, the higher temperature of the process should be used. Higher values of temperature accelerate the atoms diffusion and further phase transformations. The temperature was selected based on the analysis of the ternary aluminium-iron-silicon equilibrium system [29] (Figure 3). The temperature of 900°C was selected in order to avoid the melting point identified on the basis of the chemical composition analysis of the phases formed after the sintering process. The sintering time of 5 minutes was used on the basis of authors research [30] where the optimized structure and properties of  FeAl intermetallic phases were obtained.

Figure 3. Iron-aluminum-silicon equilibrium system with location of the obtained composite matrix material [29].

  1. Mullite ceramic only takes 20 wt% in the raw materials. Why does Al2O3 seem to occupy the major portion of the model in Fig. 1?

Response: We would like to thank the reviewer for the comment. It could be observed from Fig.4a that SiO2 particles were localised around Al2O3 particles. Thus, authors decided to present it according the observations. The main aim of the Figure 1 was to presented defragmentation of SiO2 and growth of new Al2O3 oxides.

  1. What are the very dark contrast in Fig. 2? It seems to me that those big clusters instead of the "ultrfine" Al2O3 particles, are the global feature of the obtained composite. Does it imply that the intended microstructure is not successfully obtained?

Response: We would like to thank the reviewer for the comment. Large, dark clusters are mechanically bonded SiO2 and initial Al2O3 particles from mullite and they could be further observed in Fig.4a-b. The authors did not deny the occurrence of initial Al2O3 oxides as reported in Lines 182 - 183.

  1. I don't get how the SEM micrographs reveal the "infiltration" process. Please elaborate in detail, since that is the essential part of the model.

Response: We would like to thank the reviewer for the comment. The statement  ‘A detailed analysis of the primary oxide ceramics areas revealed the initiation of metallic liquid infiltration into the porous ceramics’ in line 183 means that the infiltration was firstly observed in SiO2 areas. Wider description of SEM micrographs corelated with the model assumptions were presented in lines 185-193.

During sintering at 1030°C, the defragmentation process of primary SiO2 precipitates was observed as presented in Figure 5b. Since the liquid metal infiltrated the porous silica ceramic, the silicon atoms diffused into the FeAl3 phase. The primary two-component matrix was transformed into the ternary Al4.5FeSi (t6) phase (Figure 5c). Additionally, the oxygen atoms diffused from SiO2 ceramics and reacted with the most chemically active aluminum atoms taken from the metallic liquid. As a consequence, fine, spherical Al2O3 particles (with a diameter less than 1μm) in the iron-aluminum-silicon matrix were formed (Figure 5d). The mechanism observed for growth of a new, fine Al2O3 aluminum oxide precipitates confirmed the correctness of the microstructure evolution model.

  1. Please calculate the crystallite size of Al2O3 particles based on the XRD pattern.

Response: We would like to thank the reviewer for the comment. Halder–Wagner (HW) method was used to determine crystallite size in the function of temperature (Figure 7). The size of Al2O3 crystallites increase with the temperature and average value of 392 Å was obtained after heating to 900°C.

Figure 7. Evolution of Al2O3 crystallites size during heating

  1. Is the in situ XRD conducted on a hot stage? I suppose so. Then why is that necessary? In other word, why not collect the pattern at room temperature in the sample which is cooled down from different annealing conditions? Or, do the author believe that there will be no evolution above 900C?

Response: We would like to thank the reviewer for the comment. Authors would emphasize that:

  • The main aim of XRD was to confirm assumptions of the material model on the one hand and phase transformations occurred during sintering on the other. The in-situ measurements allow to perform such phase analysis. It has to be mentioned that IMC composites are characterized by relatively high temperature of phase transformations, thus the heating/cooling stage and further analysis at room temperature could not be precise in terms of phase structure analysis.
  • Authors used the temperature of 900°C, because aluminium content appears in the powder mixture, that is characterized by relatively low melting temperature. Authors did not want to extend a temperature above 900°C in order to avoid the liquid aluminium inside the XRD chamber.

  1. Please comment on the densification process, in addition to the chemical reaction.

Response: We would like to thank the reviewer for the comment. The comment was added to the manuscript (Lines 200 – 205) and presented below:

The SHS reaction provides a large amount of energy and subsequent immediate temperature increase. The temperature increase led to the melting of FeAl3 matrix and its transformation to Al4.5FeSi (τ6). Such material behaviour during SHS process, where the external pressure during sintering was not provided, may lead to the formation of porous structures as reported [31]. Subsequent homogenization should accelerate the densification process, and as a consequence, increase a density of the material.

Round 2

Reviewer 1 Report

The manuscript is majorly revised and the authors addressed the comments.

The conclusion still needs to be more comprehensive.

Author Response

Dear Sir or Madame,  

I would like to thank you very much for your valuable comments on our manuscript (No.: materials-832120).

Comprehensive conclusions were provided (lines 236-264), highlighted in yellow in the revised manuscript and also presented below:

The performed microstructural characterization and further analysis confirmed the correctness of the proposed model, aimed primarily at obtaining fine precipitates of Al2O3 oxide reinforcement in the intermetallic matrix of the composite. However, during the classical sintering process in the graphite dies using external pressure, the main problems were associated with a transition of silicon into the metallic liquid. This phenomenon leads to an increase of the metallic liquid phase fluidity. Such material behavior combined with external pressure, caused the loss of material during sintering process. It was proposed therefore, to perform the sintering process without external pressure. The improved sintering process allows to successfully form the composite material. Moreover, the occurrence of the assumed phase transformations was observed, in particular the defragmentation of SiO2 oxide and the construction in its primary areas of fine, spherical Al2O3 oxide precipitations, arranged in a ternary intermetallic, iron-aluminum-silicon matrix. Based on the studies performed, showing the correctness of the proposed model one can indicate it for further usage in manufacturing of IMCs composites. It should be emphasized here, that the application of external loads during the sintering process to prevent the outflow of metallic liquid infiltrating ceramics is crucial and must be taken into account. It was concluded, that a technology potentially ensuring successful sintering of IMCs, could be hot isostatic pressing (HIP).

Summing up the results of this work, one can state in particular:

  • The proposed model of composite fabrication based on the decomposition of mullite ceramics, infiltrated with a metallic liquid, allows to obtain a composite material with an intermetallic matrix reinforced with Al2O3 ceramic particles derived from the defragmentation of primary alundum ceramics and newly formed fine-grained spherical ceramics resulting from SiO2
  • The parameters of the sintering process and subsequent homogenization were significantly affected by the participation of the silicon, which significantly increase the fluidity of the metallic liquid formed during sintering and the final phase structure of the sintered matrix.
  • A correctness of the assumed model, must be supported by using a proper sintering conditions that will lead to satisfactory cohesion between the matrix and reinforcement. Future promising sintering technology seems to be hot isostatic pressing that should guarantee a fabrication of IMCs composite with the satisfactory performance.

Best regards

Mateusz Kopec

On behalf of all co-authors

Institute of Fundamental Technological Research

Polish Academy of Sciences

Reviewer 2 Report

The authors have taken all the comments seriously and addressed my concern and curiosity. I therefore suggest the acceptance of the manuscript. 

Author Response

Authors would like to thank for the reviewer’s comments on our manuscript. We appreciate reviewers valuable comments.